# WATER-GS: WATERMARK-EMBEDDED 3D GAUSSIAN SPLATTING VIA PLUG-AND-PLAY DECODER

## ABSTRACT

3D Gaussian Splatting (3DGS) has emerged as a pivotal technique for 3D scene representation, providing rapid rendering speeds and high fidelity. As 3DGS gains prominence, safeguarding its intellectual property becomes increasingly crucial since 3DGS could be used to imitate unauthorized scene creations and raise copyright issues. Existing watermarking methods for implicit NeRFs cannot be directly applied to 3DGS due to its explicit representation and real-time rendering process, leaving watermarking for 3DGS largely unexplored. In response, we propose **WATER-GS**, a novel method designed to protect 3DGS copyrights through a plug-and-play strategy. First, we introduce a pre-trained watermark decoder, treating raw 3DGS generative modules as potential watermark encoders to ensure **imperceptibility**. Additionally, we implement novel 3D distortion layers to enhance the **robustness** of the embedded watermark against common real-world distortions of point cloud data. Comprehensive experiments and ablation studies demonstrate that WATER-GS effectively embeds imperceptible and robust watermarks into 3DGS without compromising rendering efficiency and quality. Our experiments indicate that the 3D distortion layers can yield up to a 20% improvement in accuracy rate. Notably, our method is adaptable to different 3DGS variants, including 3DGS compression frameworks and 2D Gaussian splatting.

## 1 INTRODUCTION

In recent years, significant advancements in 3D scene representations have positioned 3D Gaussian Splatting (3DGS) (Kerbl et al., 2023) as a prominent methodology for 3D rendering, owing to its high fidelity and rapid rendering speeds. Accordingly, it is increasingly essential to propose a convenient and universal watermark method for 3DGS. In this case, creators can easily claim copyright for their original 3D works even if the works are widely disseminated.

Digital watermarking is a technique used to protect the intellectual property of digital assets distributed on the Internet. For other forms of digital assets (e.g., images, videos, audio, or

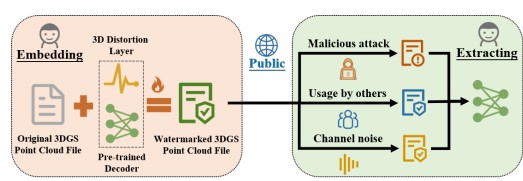

Figure 1: Before publication, creators can utilize pre-trained decoder and 3D distortion layers to embed digital signatures as watermarks into 3DGS files, thereby asserting ownership. Even when the 3DGS files undergo various distortions, the watermark can be reliably extracted, serving as effective proof of ownership.

pre-trained models), a considerable amount of work has been done on embedding invisible and robust watermarks (Zhu et al., 2018; Luo et al., 2023a; Zhang et al., 2023). In contrast, watermarking techniques related to 3D scenes are relatively scarce. Classical 3D watermarking methods primarily employ geometric properties or frequency domain transformations to embed watermarks into 3D data, such as point clouds (Ferreira & Lima, 2020) or meshes (Yoo et al., 2022). With the rise of Neural Radiance Fields (NeRFs) (Mildenhall et al., 2021), a prominent representation for 3D scenes, the embedding of watermarks has shifted to leveraging the weights of multilayer perceptrons (MLPs) that implicitly encode 3D structures and other parameters (Li et al., 2023; Song et al., 2024). However, existing 3D watermarking methods are not suitable for 3D generative scenes (3DGS), which present unique challenges and requirements due to their complex and dynamic nature. As 3DGS

are increasingly utilized in applications such as virtual reality, gaming, and digital art, the need for effective watermarking techniques becomes crucial to ensure copyright protection and provenance tracking. In light of these considerations, we propose a watermarking method specifically designed for 3DGS. Our approach not only addresses the limitations of current 3D watermarking techniques but also enhances the security and integrity of 3D generative content, thereby contributing to the broader field of digital asset protection.

Considering the unique characteristics of 3D generative scenes (3DGS), robust watermarking techniques must possess the following properties. **1): Flexibility:** The watermark should be embedded into the 3DGS parameters and extracted from the 2D rendered images. In industries where 3D technology is widely applied (e.g., gaming, filmmaking, and graphic design), users primarily access content through 2D renderings rather than 3D models. Therefore, each rendered image must carry the same copyright information. **2) Fidelity:** The watermark should be implicitly embedded within the pre-trained 3DGS file without compromising the quality of the renderings. As 3DGS is a real-time renderable representation, the entire file is uploaded online for others to download. Moreover, 3DGS files or point clouds from different frameworks may have various formats, necessitating that the watermarking method be compatible with diverse 3DGS pipelines without altering the original file format and attributes. **3) Robustness:** The watermarking method should maintain robustness, enabling the extraction of watermark information even from compressed or partially distorted 3D files. Previous NeRF watermark studies, such as (Luo et al., 2023b), have focused solely on distortions in rendered images, neglecting distortions within the NeRF network itself. We argue that distortions in the original 3D model parameters are even more critical, as watermark bits are embedded directly within them.

While imperceptibility is crucial for watermarking methods, the heightened demand for robustness distinguishes watermarking from steganography (Zhu et al., 2018). Therefore, in addition to invisibility, we prioritize enhancing the robustness of 3DGS watermarks. Instead of embedding messages in specific parameters, we propose a watermarking method that fine-tunes the entire model using a pre-trained decoder, as shown in Fig. 1. By integrating watermark messages into various parameters, the embedded messages remain consistent across different viewpoints rendered from 3DGS models. Additionally, we introduce 3D distortion layers during the fine-tuning stage to bolster watermark robustness. Previous works primarily addressed distortions in 2D renderings, neglecting the impact on point cloud files or 3DGS models. Our 3D distortion layers are designed to ensure effective watermark extraction even under severe 3D data distortions. For instance, when fine-tuned with these layers, WATER-GS improves extraction accuracy from 74.38% to 95.14% under Gaussian noise distortion applied to 3D points. In summary, the watermark extraction accuracy can reach 95% across various distortions.

Our contributions can be summarized as follows:

- We present an innovative plug-and-play watermarking method for 3DGS. This approach facilitates easy watermarking for creators without requiring additional modifications to 3DGS models, as it seamlessly integrates into various 3DGS pipelines. Our method achieves an optimal balance between robustness and imperceptibility.

- We introduce 3D distortion layers between 3DGS generation module and decoder, applying various point cloud transformations to help the model learn encodings that withstand real-world noise during transmission. Our experiments demonstrate that this training method significantly enhances watermark robustness

- We conduct extensive testing on mainstream 3DGS datasets, where our approach outperforms other baselines. Comprehensive ablation studies and analyses further validate the effectiveness of each proposed component.

## 2 RELATED WORKS

### 2.1 3D GAUSSIAN SPLATTING

In recent advancements, 3D Gaussian Splatting (3DGS) (Kerbl et al., 2023) has gained tremendous traction as a promising paradigm to 3D view synthesis, reconstructing and representing 3D scenes using millions of 3D Gaussians endowed with learnable shape and appearance attributes. Compared

to implicit representations like NeRF (Mildenhall et al., 2021), 3DGS offers remarkable acceleration in training and rendering while utilizing fewer resources and maintain high-fidelity quality. By leveraging explicit 3D Gaussian representations and differentiable tile-based rasterization (Lassner & Zollhöfer, 2020), 3DGS optimizes 3D Gaussians during training to accurately fit their local regions. This not only enhances flexibility and editability, but also enables high-fidelity, real-time rendering for 3D scene reconstruction.

In response to 3DGS's contributions to 3D representation, recent advancements and derivative works have emerged. Mip-Splatting (Yu et al., 2024) introduces a 2D Mip filter and a 3D smoothing filter to address aliasing and dilation issues, effectively eliminating high-frequency artifacts. DreamGaussian (Tang et al., 2023) presents a generative framework that incorporates UV space, significantly improving content creation efficiency. Additionally, Scaffold-GS (Lu et al., 2024) and HAC (Chen et al., 2024a) utilize anchor points to distribute local 3D Gaussians, enhancing scene coverage while reducing redundancy. The applications of 3DGS technology extend beyond its initial domain, finding success in fields like autonomous driving (Tang et al., 2023; Zhou et al., 2024), simultaneous localization and mapping (SLAM) (Yan et al., 2024; Keetha et al., 2024), multi-modal generation (Ling et al., 2024; Chen et al., 2024b), and 2D scenarios (Zhang et al., 2024). Despite its widespread adoption, research on watermarking and copyright protection in 3DGS remains limited.

## 2.2 3D DIGITAL WATERMARK

The limited popularity of 3D data has hindered the development of watermarking technologies compared to traditional media such as images, videos, and audio. Early approaches to 3D digital watermarking utilized methods like principal component analysis (PCA) (Jolliffe & Cadima, 2016), geometric techniques like analyzing vertex curvatures (Lipuš & Žalik, 2019; Praun et al., 1999; Liu et al., 2019), and frequency-domain transform approaches (Ohbuchi et al., 2002; Hamidi et al., 2019). Following the success of HiDDeN (Zhu et al., 2018), a deep image watermarking method that outperformed traditional techniques, various extensions have emerged. For instance, 3D-to-2D Watermarking(Yoo et al., 2022) adopts similar principles to address mesh watermarking problem.

Recently, NeRF-based watermarking method (Li et al., 2023; Luo et al., 2023b; Song et al., 2024; Huang et al., 2024b) have attracted increasing attention. For example, StegaNeRF (Li et al., 2023) embeds watermark information into the parameters of pre-trained NeRFs. CopyRNeRF (Luo et al., 2023b) encodes watermark messages into implicit tensors and concatenates them with color representations. NeRFProtectorr (Song et al., 2024) embeds binary messages directly during the creation of NeRFs. However, watermarking for emerging 3DGS has yet to be explored. Our work pioneers robust watermarking for 3DGS by utilizing the 3DGS generative module as an watermark encoder, allowing creators to embed watermark information both effectively and imperceptibly.

**3D Gaussian Splatting (3DGS).** (Kerbl et al., 2023) represents the scene through numerous Gaussians, which are rendered from different viewpoints using differentiable splatting and tile-based rasterization. Each Gaussian is initialized from structure-from-motion (SfM) (Schonberger & Frahm, 2016) and characterized by a 3D covariance matrix $\boldsymbol{\Sigma} \in \mathbb{R}^{3\times3}$ and location $\boldsymbol{\mu} \in \mathbb{R}^3$ (mean),

$$\mathcal{G}(\boldsymbol{x}) = \exp(-\frac{1}{2}(\boldsymbol{x} - \boldsymbol{\mu})^\top \boldsymbol{\Sigma}^{-1}(\boldsymbol{x} - \boldsymbol{\mu})), \tag{1}$$

where $\boldsymbol{x} \in \mathbb{R}^3$ denotes the position of a random point, and $\boldsymbol{\Sigma}$ can be decomposed into a scaling matrix $\boldsymbol{S} \in \mathbb{R}^{3\times3}$ parameterized by $\boldsymbol{s} \in \mathbb{R}^3$ and a rotation matrix $\boldsymbol{R} \in \mathbb{R}^{3\times3}$ parameterized by $\boldsymbol{r} \in \mathbb{R}^4$. $\boldsymbol{\Sigma}$ is a positive semi-definite matrix which can be expressed as $\boldsymbol{\Sigma} = \boldsymbol{R}\boldsymbol{S}\boldsymbol{S}^\top \boldsymbol{R}^\top$, $\boldsymbol{S}$ is a diagonal matrix. To render an image from a specific viewpoint, an efficient 3D-to-2D Gaussian mapping (Zwicker et al., 2001) projects the Gaussians onto the 2D image plane, rendering pixel values through $\alpha$-composed blending (Kopanas et al., 2022). Let $\boldsymbol{C} \in \mathbb{R}^{H\times W\times3}$ represent the color of the rendered image. The rendering process is outlined as follows:

$$\boldsymbol{C}[\boldsymbol{p}] = \sum_{i\in I} \boldsymbol{c}_i \sigma_i \prod_{j=1}^{i-1}(1-\sigma_j), \quad \sigma_i = \alpha_i \exp(-\frac{1}{2}(\boldsymbol{p} - \hat{\boldsymbol{\mu}})^\top \hat{\boldsymbol{\Sigma}}^{-1}(\boldsymbol{p} - \hat{\boldsymbol{\mu}})), \tag{2}$$

where $\boldsymbol{c} \in \mathbb{R}^3$ denotes the view-dependent color, modeled by Spherical Harmonic (SH) coefficients $\boldsymbol{h} \in \mathbb{R}^{3\times(k+1)^2}$, $k$ is the order of the spherical harmonics. $I$ is the set of sorted Gaussians contribut-

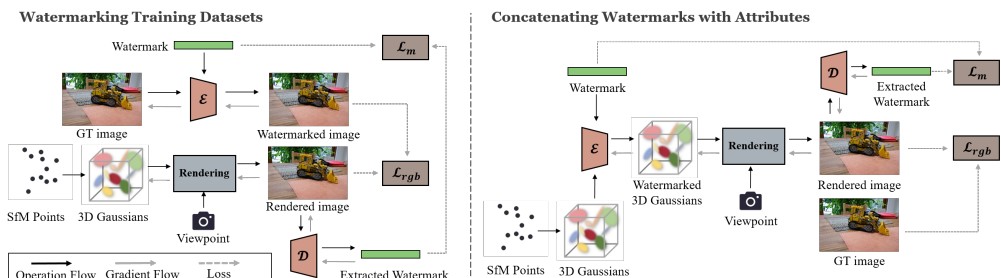

Figure 2: Two intuitive solutions for 3DGS watermarking. **Watermarking Training Datasets** draws inspiration from existing model watermarking techniques, leveraging watermarked datasets to embed watermarks. **Concatenating Watermarks with Attributes** incorporates an explicit encoder to seamlessly integrate watermarks with specific attributes.

ing to the rendered pixel $p$. $\alpha \in \mathbb{R}^1$ measures the opacity of each Gaussian after 2D projection. $\hat{\mu}$ and $\hat{\Sigma}$ represent the 2D mean position and covariance of the projected 3D Gaussian, respectively.

## 3 METHOD

**Problem Setup.** NeRF-based watermark methods (Luo et al., 2023b; Song et al., 2024) typically embed messages within model weights, allowing extraction from renderings at various viewpoints. In contrast, 3DGS utilizes an explicit 3D representation, where each point's attributes carry clear physical meanings. Furthermore, the entire point cloud files of trained 3DGS are uploaded online, *meaning that the watermark embedding process must not alter the structure of the 3DGS file or the format of existing attributes.* Therefore, we assume that a complete (or partially distorted, as long as it doesn't affect the rendered images) 3DGS file is accessible during the watermark extraction stage, enabling extraction from each viewpoint's rendering. Creators can either download a pre-trained decoder from the cloud or train one themselves. In addition to the proposed WATER-GS, we explore other intuitive solutions as our baselines illustrated in Fig. 2.

**Watermarking Training Datasets.** Inspired by previous work on generative model watermarking (Yu et al., 2021; Zhao et al., 2023), we propose embedding binary strings within training images using a watermark encoder before training the 3DGS. Furthermore, we aim to explore whether the resulting renderings retain the same quality when the training images undergo 2D watermarking.

**Concatenating Watermarks with Attributes.** Traditional deep watermarking methods design encoders to extract deep features from watermark messages and cover images. These features are then concatenated with the raw cover image to generate a watermarked image. CopyRNeRF (Luo et al., 2023b) propose transforming secret watermark messages into higher dimensions and fusing them with the spatial information and color representations of NeRFs. Furthermore, we aim to explore whether an encoder can align watermark messages with specific attributes of 3DGS, such as the spherical harmonic coefficients $h$, and extract these messages from 2D renderings.

As described in Section 3, both solutions mentioned above incorporate an additional encoder to embed watermarks. However, experiments in Section 4.3 demonstrate that neither approach successfully extracts watermarks from 2D renderings. Rather than explicitly introducing a new encoder, we propose that the original 3DGS generative network can serve as a potential encoder. Figure 3 illustrates the overall processing pipeline of the proposed WATER-GS method. Our approach aims to embed $l$-bit binary messages $m \in \{0, 1\}^l$ within the 3DGS file prior to publication. In the following sections, we first outline the process for obtaining a universal decoder in Section 3.1. We then describe the method for embedding watermarks within 3DGS and discuss the roles of the 3D distortion layers in Section 3.2.

### 3.1 PLUG-AND-PLAY WATERMARK DECODER

The core of our approach involves integrating an external decoder that can effectively disentangle messages from 2D renderings within the 3DGS pipeline. This decoder features a plug-and-play ca-

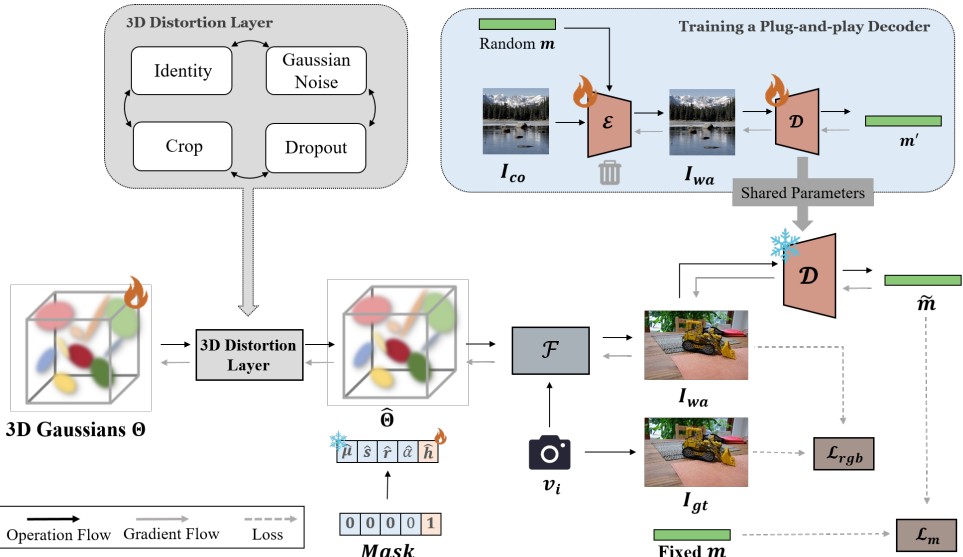

Figure 3: Illustration of our WATER-GS framework. **(1) Training a plug-and-play decoder.** In this stage, a message encoder $\mathcal{E}$ and a decoder $\mathcal{D}$ are trained end-to-end. **(2) Fine-tuning the 3D Gaussians.** In this stage, creators fine-tune the original 3D Gaussians using the pre-trained decoder to embed fixed messages. They have the flexibility to control which parameters are frozen by employing a self-defined mask. **(3) Distorstion layers.** The 3D distortion layers $\mathcal{N}$ introduce alterations to the 3D Gaussians $\Theta$, resulting in a distorted $\hat{\Theta}$.

pability, enabling seamless incorporation into the existing 3DGS framework. Consequently, creators can embed watermarks with minimal modifications to its training and rendering processes.

In this paper, we introduce a novel end-to-end decoder training framework inspired by HiDDeN (Zhu et al., 2018). Our approach innovatively integrates the watermark encoder network $\mathcal{E}$ and the decoder network $\mathcal{D}$ into a single optimized pipeline using the COCO dataset (Lin et al., 2014). During the optimization process, we input a cover image $\boldsymbol{I}_{co}$ alongside a watermark message $\boldsymbol{m} \in \{0,1\}^l$ into the encoder $\mathcal{E}$. The encoder produces the watermarked image $\boldsymbol{I}_{wa}$, which is then processed by the decoder $\mathcal{D}$ to extract the messages $\boldsymbol{m}' \in \{0,1\}^l$. This framework represents a significant advancement in watermarking techniques, as it allows for simultaneous optimization of both embedding and extraction processes. The training procedure is outlined as follows:

$$\boldsymbol{I}_{wa} = \mathcal{E}(\boldsymbol{I}_{co}, \boldsymbol{m}), \quad \boldsymbol{m}' = \mathcal{D}(\boldsymbol{I}_{wa}). \tag{3}$$

Finally, the whole network is optimized by minimizing the Mean Squared Error (MSE) loss between images and the Binary Cross Entropy (BCE) loss between messages as:

$$\mathcal{L}_i = \mathrm{MSE}(\boldsymbol{I}_{co}, \boldsymbol{I}_{wa}), \quad \mathcal{L}_m = \mathrm{BCE}(\boldsymbol{m}, \boldsymbol{m}'). \tag{4}$$

After training, the decoder $\mathcal{D}$ learns the embedding pattern, enabling the 3DGS generative model to incorporate this knowledge during the fine-tuning process.

### 3.2 ROBUST WATERMARK EMBEDDING

**Rendering watermarked images.** The raw 3DGS generative networks naturally serve as effective message embedding encoders due to their robust feature fusion and representation capabilities. Consequently, our method eliminates the need for an additional encoder to explicitly generate watermarked images. Instead, we draw inspiration from fixed neural networks (FNNs) (Kishore et al., 2021), which conceptualize image steganography as the addition of adversarial perturbations to cover images. Given a fixed decoder $\mathcal{D}$ and a watermark $\boldsymbol{m} \in \{0,1\}^l$, the FNNs produce the watermarked image $\boldsymbol{I}_{wa}$ by applying a trained perturbation $\Delta$ to the cover image $\boldsymbol{I}_{co}$ as follows:

$$\boldsymbol{I}_{wa} = \boldsymbol{I}_{co} + \Delta, \quad \boldsymbol{m}' = \mathcal{D}(\boldsymbol{I}_{wa}). \tag{5}$$

---

**Algorithm 1** Fine-tuning the 3DGS for watermark embedding

---

    **Models**: decoder $\mathcal{D}$, 3DGS model $\mathcal{F}$ with parameter $\Theta$, 3D distortion layers $\mathcal{N}$
    **Data**: watermark $\boldsymbol{m}$, extracted watermark $\tilde{\boldsymbol{m}}$, training images $\{I_i\}$ with viewpoints $\{v_i\}$, rendered image $I_{pred}$
    **Hyper-parameters**: mask $\boldsymbol{M}$, learning rate $\eta$, optimization steps $n$
    **Output**: Watermarked 3DGS model $\mathcal{F}$ with parameter $\tilde{\Theta}$

1:  $\tilde{\Theta} \leftarrow \Theta$
2:  **for** $n$ iterations **do**
3:     Randomly sample a training pose $v_i$
4:     $\boldsymbol{I}_{pred} = \mathcal{F}(v_i, \mathcal{N}(\tilde{\Theta}))$
5:     $\tilde{\boldsymbol{m}} = \mathcal{D}(\boldsymbol{I}_{pred})$
6:     Compute message loss $\mathcal{L}_m$ and standard loss $\mathcal{L}_{rgb}$ as in Eq. (8), (9)
7:     Update $\tilde{\Theta}$ with $\eta \cdot (\frac{\partial \mathcal{L}_{tot}}{\partial \tilde{\Theta}} \odot \boldsymbol{M})$
    **return** $\tilde{\Theta}$

---

Our watermark embedding algorithm is outlined in Algorithm 1. In our approach, we propose fine-tuning the 3DGS model to render a watermarked image $\tilde{\boldsymbol{I}}$ instead of a standard image $\boldsymbol{I}$. To maintain plug-and-play compatibility, we avoid making additional modifications to the original 3DGS pipeline $\mathcal{F}$ with parameters $\Theta = \{\boldsymbol{\mu}, \boldsymbol{s}, \boldsymbol{r}, \alpha, \boldsymbol{h}\}$. We first train $\mathcal{F}$ following the regular process until the quality of the rendered image $\boldsymbol{I}$ stabilizes. Subsequently, we fine-tune $\Theta$ with the fixed $\mathcal{D}$ to obtain $\tilde{\Theta} = \{\tilde{\boldsymbol{\mu}}, \tilde{\boldsymbol{s}}, \tilde{\boldsymbol{r}}, \tilde{\alpha}, \tilde{\boldsymbol{h}}\}$, which renders the watermarked images $\tilde{\boldsymbol{I}}$. Creators have the option to fine-tune either the entire or a partial $\Theta$ using a self-defined mask $\boldsymbol{M}$. The process proceeds as follows, in accordance with Eq. (2):

$$\tilde{\boldsymbol{m}} = \mathcal{D}(\tilde{\boldsymbol{I}}), \quad \tilde{\boldsymbol{I}}[\boldsymbol{p}] = \sum_{i \in I} \tilde{\boldsymbol{c}}_i \tilde{\sigma}_i \prod_{j=1}^{i-1}(1 - \tilde{\sigma}_j) \tag{6}$$

**3D distortion layers.** Previous research on 3D-based digital watermarking (Yoo et al., 2022; Luo et al., 2023b; Song et al., 2024) has primarily focused on distortions in 2D images, such as cropping and JPEG compression, while overlooking the distortions inherent in 3D point cloud files that can arise from transmission or malicious alterations. As a result, even though rendered images exhibit robustness against various 2D distortions, compromised 3D files may not produce accurate watermarked images. To bridge this gap, we introduce 3D distortion layers $\mathcal{N}$, defined as follows:

$$\hat{\Theta} = \mathcal{N}(\tilde{\Theta}), \quad \mathcal{N} = \{\text{Identity}, \text{GN}, \text{Dropout}, \text{Crop}\}. \tag{7}$$

This innovative approach enables our framework to effectively account for distortions specific to 3D data, enhancing the reliability of watermark extraction. The Identity layer maintains $\tilde{\Theta}$ unchanged to ensure basic extraction capability in the absence of distortion. The Gaussian Noise (GN) layer applies a Gaussian kernel with width $\sigma$ to blur $\tilde{\Theta}$, thereby enhancing robustness against minor parameter variations. The Dropout layer randomly removes a fraction $p$ of the points, increasing resilience against specific pruning compressions. Lastly, the Crop layer eliminates consecutive intervals corresponding to a fraction $p$ to improve robustness against partial data loss.

Given that 3DGS lacks standardized compression methods such as JPEG, leading to a variety of point cloud file formats, we deliberately exclude compression from the 3D distortion layers. However, robustness against specific compression techniques can still be attained by incorporating a plug-and-play decoder within the 3DGS compression pipeline (Lee et al., 2024; Niedermayr et al., 2024). This integration allows our framework to adaptively respond to diverse compression challenges, thereby enhancing the overall reliability of watermark extraction.

**Training details.** The training set is $\{\boldsymbol{I}_{gt}^{(i)}\}^n$ and the watermark message is $\boldsymbol{m} \in \{0, 1\}^l$. The learnable parameters in fine-tuning stage are $\Theta = \{\boldsymbol{\mu}, \boldsymbol{s}, \boldsymbol{r}, \alpha, \boldsymbol{h}\}$. After training, we can obtain the watermarked image set $\{\boldsymbol{I}_{wa}^{(i)}\}^n$ rendered by $\tilde{\Theta}$. During optimization, we minimize the BCE loss between messages and adhere to the original constraints of the original 3DGS pipeline:

$$\mathcal{L}_m = \text{BCE}(\boldsymbol{m}, \tilde{\boldsymbol{m}}), \quad \tilde{\boldsymbol{m}} = \mathcal{D}(\boldsymbol{I}_{wa}) \tag{8}$$

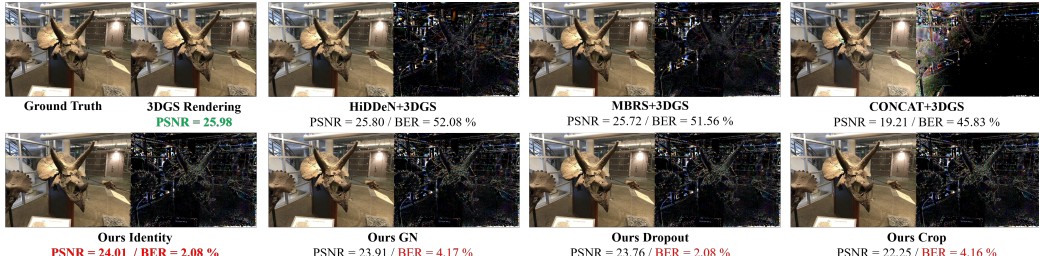

Figure 4: Visual quality comparisons of each baseline. We display both the rendered images and the corresponding residual images ($\times$ 10). WATER-GS demonstrates the optimal balance between rendering quality and watermark extraction accuracy.

$$\mathcal{L}_{rgb} = (1 - \beta) \cdot \mathcal{L}_1(\boldsymbol{I}_{gt}, \boldsymbol{I}_{wa}) + \beta \cdot \mathcal{L}_{SSIM}(\boldsymbol{I}_{gt}, \boldsymbol{I}_{wa}), \tag{9}$$

where $\beta$ denote the balancing weight of $\mathcal{L}_1$ and $\mathcal{L}_{SSIM}$. The total is $\mathcal{L}_{tot} = \mathcal{L}_{rgb} + \gamma \cdot \mathcal{L}_m$, where $\gamma$ is utilized to balance the optimization between robustness and invisibility of watermark embedding. During the fine-tuning process, we implement an adaptive density control strategy to facilitate the splitting and merging of Gaussian points, analogous to the approach used in 3DGS.

## 4 EXPERIMENTS

In this section, we first detail the implementation of our framework, followed by comprehensive experiments to evaluate the performance of our model. Additionally, we present a visual comparison of the rendered images before and after watermark embedding. Finally, we conduct ablation studies to explore the impact of watermark embedding positions within the 3DGS model and assess the effectiveness of noise layers.

### 4.1 EXPERIMENTAL SETUP

**Datasets.** We conduct experiments on real-world datasets: LLFF (Mildenhall et al., 2019), Mip-NeRF360 (Barron et al., 2021) and Tanks&Temples (Knapitsch et al., 2017). We choose the train and truck scenes from the Tanks&Temples dataset, similar to the (Kerbl et al., 2023). We present average values across all testing viewpoints in our experiments.

**Evaluation Metrics.** We evaluate the performance of our watermarking method from two perspectives: robustness and imperceptibility. For robustness, we utilize the Bit Error Rate (BER) metric to assess the extraction accuracy of the watermark message. To evaluate imperceptibility, we employ metrics such as PSNR, MS-SSIM and VGG-LPIPS (Simonyan & Zisserman, 2014; Zhang et al., 2018), which measure the distortion between rendered images from the raw and watermarked 3DGS model.

**Implementation Details.** Our method fine-tunes the model for 10K to 30K iterations to embed the watermark into pre-trained scene. The entire fine-tuning process takes approximately half to multiple hours according to the size of scenes. The watermark decoder utilizes the HiDDeN (Zhu et al., 2018) structure, trained on the COCO (Lin et al., 2014) dataset. Experiments are conducted using NVIDIA GeForce RTX 3090 GPU and PyTorch.

**Baselines.** We compare our WATER-GS with three baselines for a fair assessment: 1) **"HiDDeN + 3DGS"**, which involves training the 3DGS model with watermarked images processed by HiDDeN (Zhu et al., 2018); 2) **"MBRS + 3DGS"**, which uses watermarked images processed by the SOTA image watermark method MBRS (Jia et al., 2021); 3) **"CONCAT + 3DGS"**, which concatenates the watermark message with spherical harmonic (SH) coefficients, ensuring minimal impact on image rendering.

### 4.2 QUALITATIVE RESULTS AND ANALYSIS

We first compare the reconstruction quality against all baselines, with results presented in Fig. 4. The residual images between the rendered outputs and the ground truth have been magnified ten

Table 1: Quantitative results of robustness performance under various 3DGS distortions. "Identity" signifies no applied distortion. The results are averaged on the selected dataset scenes. Our proposed method achieves the best performance across all settings.

| Dataset | Method | Bit Error Rate (%) ↓ | | | |
|---|---|---|---|---|---|
| | | Identity | GN | Dropout | Crop |
| LLFF | HiDDeN + 3DGS | 47.50 | 48.13 | 47.66 | 47.43 |
| | MBRS + 3DGS | 50.71 | 50.75 | 50.57 | 50.75 |
| | CONCAT+3DGS | 43.13 | 43.25 | 42.95 | 42.95 |
| | Ours | **3.26** | **4.86** | **3.93** | **4.32** |
| MIP-NeRF360 | HiDDeN + 3DGS | 53.94 | 53.83 | 53.93 | 53.99 |
| | MBRS + 3DGS | 50.78 | 50.81 | 50.75 | 50.79 |
| | CONCAT + 3DGS | 59.84 | 59.88 | 59.83 | 59.93 |
| | Ours | **9.22** | 20.77 | 10.09 | 10.77 |
| Tanks&Temples | HiDDeN + 3DGS | 47.92 | 48.02 | 47.92 | 47.86 |
| | MBRS + 3DGS | 48.40 | 48.48 | 48.44 | 48.28 |
| | CONCAT + 3DGS | 50.73 | 50.68 | 50.52 | 50.73 |
| | Ours | **6.30** | 29.11 | **8.59** | **8.90** |

times for enhanced visibility. With the exception of the "CONCAT + 3DGS" method, all other approaches achieve high reconstruction quality. Notably, our method exhibits the lowest bit error rate (BER) among all watermark embedding techniques, incurring minimal sacrifices in image quality. Furthermore, despite the distortions present in the rendering point cloud, watermarks can still be accurately extracted from the rendered images.

**Watermarking Training Datasets.** For methods that aim to transfer watermarks from training images to rendered images, although they achieve nearly 100% extraction accuracy on training datasets, their bit error rates (BER) are significantly high for 3DGS rendered images. The elevated error rates observed in "HiDDeN + 3DGS" and "MBRS + 3DGS" demonstrate that the watermark cannot withstand the training of the 3DGS generative model. As illustrated in Fig. 5, we present a comparison of the residual images before and after rendering, showing that the rendering process disrupts the watermark embedding pattern, resulting in extraction failures. Consequently, training the 3DGS with watermarked images does not allow the 3DGS network to effectively learn the watermark pattern.

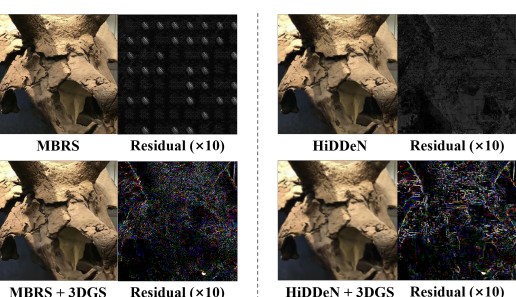

Figure 5: A visual comparison of the watermark patterns pre- and post-rendering. The residual images indicate that the watermark pattern is disrupted.

**Concatenating Watermarks with Attributes.** Methods that employ an explicit encoder to concatenate watermarks with existing attributes significantly degrade the visual quality of rendered images. Since each point's attributes in the 3DGS point cloud file possess clear physical meanings, we avoid embedding watermarks in parameters related to position. Instead, we embed the watermark within the spherical harmonics (SH) coefficients $h$, which define color. However, the results indicate that any explicit modifications to existing attributes can severely compromise the quality of rendered images, rendering invisible watermark embedding impossible. A detailed analysis is provided Appendix C.1.

### 4.3 QUANTITATIVE RESULTS AND ANALYSIS

We conducted quantitative experiments using 48-bit messages, allowing for a high watermark capacity for 3D data (Luo et al., 2023b; Song et al., 2024). Tab. 1 presents the Bit Error Rate (BER) across various point cloud distortions, demonstrating that our method consistently achieves the highest extraction accuracy compared to all baselines. The "HiDDeN + 3DGS" and "MBRS + 3DGS"

Table 2: Quantitative results of extraction accuracy and reconstruction quality. "w/o DL" indicates fine-tuning without distortion layers. Note that the quality of raw 3DGS rendered images is inferior to that of NeRF. Red text indicates performance degradation, while green text signifies performance improvement.

| Method | BER (%) ↓ | PSNR ↑ | SSIM ↑ | LPIPS ↓ |
|---|---|---|---|---|
| NeRF | N/A | 30.38 | 0.952 | 0.036 |
| CopyRNeRF | 8.84 | 25.80 / 4.58 ↓ | 0.830 / 0.122 ↓ | 0.103 / 0.067 ↑ |
| 3DGS | N/A | 24.99 | 0.801 | 0.175 |
| Ours w/o DL | **5.10** | 23.35 / 1.64 ↓ | 0.796 / 0.005 ↓ | 0.188 / 0.001 ↑ |
| Ours | **2.83** | 22.77 / 2.22 ↓ | 0.802 / 0.001 ↑ | 0.190 / 0.015 ↑ |

Table 3: Ablation studies on the proposed 3D distortion layers. "w/o DL" indicates fine-tuning without the 3D distortion layers. The watermark length is set to 48 bits.

| Method | Identity | | Gaussian Noise | | Dropout | | Crop | |
|---|---|---|---|---|---|---|---|---|
| | BER | PSNR | BER | PSNR | BER | PSNR | BER | PSNR |
| Ours w/o DL | 4.82% | 21.95 | 25.62% | 21.52 | 12.24% | 21.29 | 11.82% | 21.25 |
| Ours | **3.26%** | 21.83 | **4.86%** | **21.80** | **3.93%** | **21.65** | **4.32%** | **21.63** |

methods completely fail in watermark embedding, as a "BER $\geq 50\%$" indicates random guessing during the extraction process. The "CONCAT + 3DGS" method shows some effectiveness in simple scenes from the LLFF (Mildenhall et al., 2019) dataset but fails to perform adequately in more complex scenes.

Additionally, we compared our method to the NeRF-based watermarking approach CopyRNeRF (Luo et al., 2023b), conducting experiments under optimal settings for CopyRNeRF using 16-bit messages. As shown in Tab. 2, our method outperformed CopyRNeRF in extraction accuracy in both scenarios, with and without the 3D distortion layers. Furthermore, a fair comparison of image quality is challenging, as the quality of raw 3DGS rendered images is generally inferior to that of NeRF. However, considering the specific decline values of the indicators, our method demonstrates less performance deterioration. Notably, experiments with CopyRNeRF indicated that 48 bits represents the upper bound capacity, suggesting that the 3DGS format can offer superior coverage compared to NeRF.

## 4.4 ABLATION STUDY

**3D Distortion Layers.** Ablation studies on 3D distortion layers are presented in Tab. 3. We observe that the 3DGS fine-tuned without 3D distortion layers performs poorly when subjected to various distortion types. Among these distortions, Gaussian noise has the most detrimental effect on the watermark, while the distortion layers enhance extraction accuracy by **20.76%**. The low BER reported in Tab. 3 indicates that models can develop robustness to a range of 3D distortions when these are incorporated into the fine-tuning process. Furthermore, we observe improvements in the quality of images rendered from distorted 3DGS files when utilizing the 3D distortion layers.

**Embedding Positions.** Ablation studies on various fine-tuning methods of the 3DGS are presented in Tab. 4. As described in Sec 2.2, the 3DGS file comprises $\boldsymbol{\Theta} = \{\boldsymbol{\mu}, \boldsymbol{s}, \boldsymbol{r}, \alpha, \boldsymbol{h}\}$ and position parameters $\boldsymbol{xyz}$. The coefficients $\boldsymbol{h}$ consist of two parts: 0-order $\boldsymbol{h}_{dc}$, which determines ambient and diffuse light; and the 1nd, 2nd and 3rd orders $\boldsymbol{h}_{rest}$, which govern specular light. The rotation

Table 4: Ablation studies on different watermark embedding positions of the proposed WATER-GS. The watermark length is set to 48 bits.

| Embedding Positions | BER (%) ↓ | PSNR ↑ | SSIM ↑ | LPIPS ↓ |
|---|---|---|---|---|
| $\boldsymbol{xyz}$ | 39.10 | 23.61 | 0.773 | 0.211 |
| $\boldsymbol{h}_{dc}$ | 13.03 | 23.95 | 0.783 | 0.206 |
| $\boldsymbol{h}_{rest}$ | 10.47 | **24.46** | 0.788 | 0.199 |
| ALL | **5.10** | 22.85 | 0.769 | 0.188 |

Table 5: Ablation studies on various 3DGS framework. Notice that the compression process of Compact3D (Lee et al., 2024) compromises the integrity of the point cloud file and watermark pattern. The watermark length is set to 48 bits.

| Method | BER (%) ↓ | PSNR ↑ | SSIM ↑ | LPIPS ↓ |
|---|---|---|---|---|
| 3DGS | N/A | 24.99 | 0.801 | 0.175 |
| 3DGS + WATERGS | 5.10 | 23.35 | 0.796 | 0.188 |
| 2DGS | N/A | 24.89 | 0.812 | 0.189 |
| 2DGS + WATERGS | 8.53 | 21.49 | 0.765 | 0.270 |
| Compact3D | N/A | 24.53 | 0.792 | 0.189 |
| Compact3D + WATERGS | 9.85 | 21.70 | 0.743 | 0.275 |

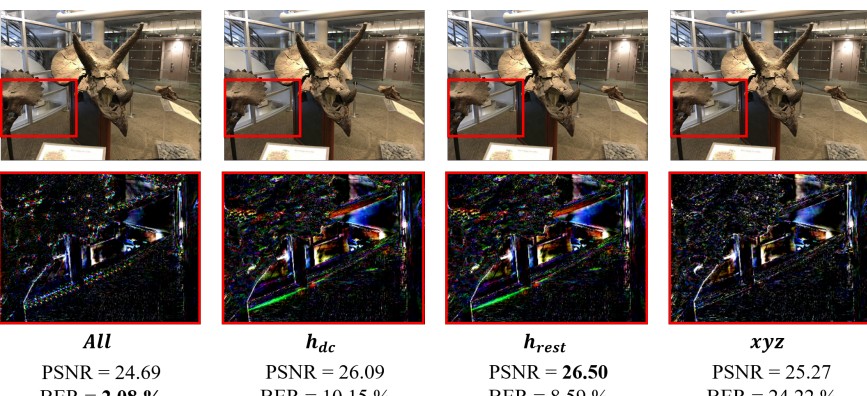

|   |   |   |   |
|---|---|---|---|
| *All* | $h_{dc}$ | $h_{rest}$ | $xyz$ |
| PSNR = 24.69 | PSNR = 26.09 | PSNR = **26.50** | PSNR = 25.27 |
| BER = **2.08 %** | BER = 10.15 % | BER = 8.59 % | BER = 24.22 % |

Figure 6: The rendered images and localized detailed residual images (×10) different watermark embedding positions.

factor $r$, scaling factor $s$ and opacity factor $\alpha$ are activated by the normalization layer, exponential layer and Sigmoid layer, respectively. Consequently, these three factors cannot be fine-tuned independently. Instead, we conduct experiments to separately fine-tune the position parameters $xyz$ and SH coefficients $h$. As shown in Fig. 6, fine-tuning the parameter $h_{dc}$ or $h_{rest}$ results in color distortion in the image, while solely fine-tuning $xyz$ does not enable full watermark embedding. Therefore, fine-tuning all parameters achieves the optimal balance between extraction accuracy and reconstruction quality. Additional qualitative comparisons can be found in Appendix C.2.

**Adaption to diverse 3DGS variants.** In our default configuration, we utilize the original 3D Gaussian Splatting (3DGS) pipeline (Kerbl et al., 2023). However, WATER-GS is readily adaptable to various 3DGS variants. As illustrated in Tab. 5, we further integrate the decoder with 2D Gaussian Splatting (2DGS) (Huang et al., 2024a) and the Compact3D compression pipeline (Lee et al., 2024). The results demonstrate that the watermark can be accurately extracted from 2DGS. In the case of Compact3D, the watermark exhibits robustness developed during the fine-tuning stage, achieving a 90.15% extraction accuracy even when point cloud files are compressed.

## 5 CONCLUSION

In this paper, we present a novel plug-and-play approach for safeguarding the copyright of 3D Gaussian Splatting (3DGS). Our method leverages a universal decoder, enabling creators to embed unique signatures as watermarks into variants 3DGS models, which can be accurately extracted from rendered images captured from any viewpoints. To enhance the robustness of the embedded watermarks, we incorporate 3D distortion layers into the fine-tuning process, effectively mitigating potential distortions encountered during rendering. To the best of our knowledge, WATER-GS is the first method to address robust watermarking for 3DGS while pioneering the application of distortion layers in 3DGS point cloud data. Experimental results validate the superiority of our approach and demonstrate its seamless integration into diverse 3DGS variants. Future work will focus on further enhancing the visual quality of watermarked 3DGS models and improving robustness under complex real-world conditions.

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

## A IMPLEMENTATION DETAILS

### A.1 HYPER-PARAMETERS

**Decoder network architecture.** We use the network architecture proposed by HiDDeN (Zhu et al., 2018). The encoder $\mathcal{E}$ consists of four stacked layers of convolution followed by ReLU activation functions and the decoder $\mathcal{D}$ consists of seven stacked layers of convolution followed by ReLU activation functions. The middle channels are set to 64. During training stage, the images are cropped to $256 \times 256$ and the model is trained for 300 epochs.

**Distortion layer strength factors.** For Gaussian noise, we add noise to position parameters $\boldsymbol{xyz}$, the kernel width $\sigma$ is set to 0.01. For Dropout and Crop noise, we drop a percentage $p = 10\%$ of points from original 3DGS parameters $\boldsymbol{\Theta}$. Creators can fine-tune this factors to change the strength of distortion. Fine-tuning the model with stronger distortion degrees can enhance robustness of watermark while sacrifice the image quality.

**Loss weights.** The total loss $\mathcal{L}_{tot}$ is consist of $\mathcal{L}_{rgb}$ and $\mathcal{L}_m$, balanced by $\gamma$. We prefer to remain hyper-parameters of raw frameworks unchanged and control the $\gamma$ to fit different scenes and datasets. In summary, we use a smaller $\gamma$ for the fewer points 3DGS. In this paper, we set $\gamma = 0.1$ for LLFF (Mildenhall et al., 2019) dataset and $\gamma = 0.3$ for others.

**Fine-tuning iterations.** Excessive training epochs can degrade image quality; therefore, we set the training duration to range between 10,000 and 30,000 iterations. When the bit error rate (BER) decreases to 0.0%, we implement early stopping of the fine-tuning process.

### A.2 TIMING

Tab. 6 presents the average time required to train models for 10K iterations across various 3DGS variants and datasets on a NVIDIA GeForce RTX 3090 GPU. We also investigated the use of watermark messages of varying lengths and found that the fine-tuning time remains approximately constant. Incorporating the 3D distortion layers results in a slowdown of the training speed due to the relatively slow tensor replacement operations. Nevertheless, our method achieves a training time of less than 1.5 hours, outperforming NeRF-based methods, which typically require multiple hours for completion.

Table 6: Time in minutes for different 3DGS variants and datasets.

| Method | Time in minutes | | |
|---|---|---|---|
| | LLFF | Mip-NeRF360 | Tanks&Temples |
| 3DGS | 4 | 6 | 4 |
| 3DGS + WATER-GS w/o DL | 17 | 60 | 30 |
| 3DGS + WATER-GS | 50 | 100 | 60 |
| 2DGS | 15 | 20 | 18 |
| 2DGS + WATER-GS w/o DL | 15 | 20 | 18 |
| 2DGS + WATER-GS | 45 | 65 | 45 |
| Compact3D | 5 | 6 | 5 |
| Compact3D + WATER-GS w/o DL | 30 | 65 | 35 |
| Compact3D + WATER-GS | 55 | 100 | 55 |

# B ADDITIONAL QUANTITATIVE EXAMPLES

## B.1 QUANTITATIVE RESULTS FOR MORE BIT LENGTHS

We examined the relationship between BER and four different watermark capacities, revealing that increasing watermark bits has minimal impact on extraction accuracy. This stability arises because our method fine-tunes the entire 3DGS structure for implicit watermark embedding, rather than relying on specific parameter domains. Tab. 7, 8, 9 illustrate the performance of different watermark lengths across the three datasets, respectively.

Table 7: Relationship between BER and watermark capacities. We show the results on dataset LLFF (Mildenhall et al., 2019). Note that the 0.00 number is exactly zero.

| Bit Number | BER (%) ↓ | PSNR ↑ | SSIM ↑ | LPIPS ↓ |
|---|---|---|---|---|
| 0 bit | N/A | 24.99 | 0.801 | 0.175 |
| 8 bits | 0.00 | 25.17 | 0.805 | 0.181 |
| 16 bits | 2.83 | 22.77 | 0.802 | 0.190 |
| 32 bits | 5.45 | 25.11 | 0.803 | 0.185 |
| 48 bits | 3.26 | 21.83 | 0.769 | 0.258 |

Table 8: Relationship between BER and watermark capacities. We show the results on dataset Mip-NeRF360 (Barron et al., 2021).

| Bit Number | BER (%) ↓ | PSNR ↑ | SSIM ↑ | LPIPS ↓ |
|---|---|---|---|---|
| 0 bit | N/A | 27.52 | 0.814 | 0.224 |
| 8 bits | 8.88 | 25.17 | 0.823 | 0.353 |
| 16 bits | 8.12 | 23.20 | 0.768 | 0.359 |
| 32 bits | 10.89 | 24.79 | 0.866 | 0.307 |
| 48 bits | 12.22 | 24.75 | 0.761 | 0.363 |

Table 9: Relationship between BER and watermark capacities. We show the results on dataset Tanks&Temples (Knapitsch et al., 2017).

| Bit Number | BER (%) ↓ | PSNR ↑ | SSIM ↑ | LPIPS ↓ |
|---|---|---|---|---|
| 0 bit | N/A | 23.90 | 0.844 | 0.183 |
| 8 bits | 5.35 | 21.71 | 0.765 | 0.303 |
| 16 bits | 6.56 | 19.30 | 0.734 | 0.352 |
| 32 bits | 5.86 | 22.41 | 0.783 | 0.270 |
| 48 bits | 6.30 | 19.96 | 0.768 | 0.308 |

## C ADDITIONAL QUALITATIVE EXAMPLES

### C.1 QUALITATIVE RESULTS FOR CONCATENATING WATERMARKS WITH ATTRIBUTES

We concatenate watermark messages within the SH coefficients $h$, which define color and have minor impact on image quality. 0-order $h_{dc}$ determines ambient and diffuse light. The 1nd, 2nd and 3rd orders $h_{rest}$ govern specular light. As illustrated in Fig. 7, changing $h_{dc}$ influences the structure and coarse feature of rendered images, while changing $h_{rest}$ affects reflection and fine-grained details.

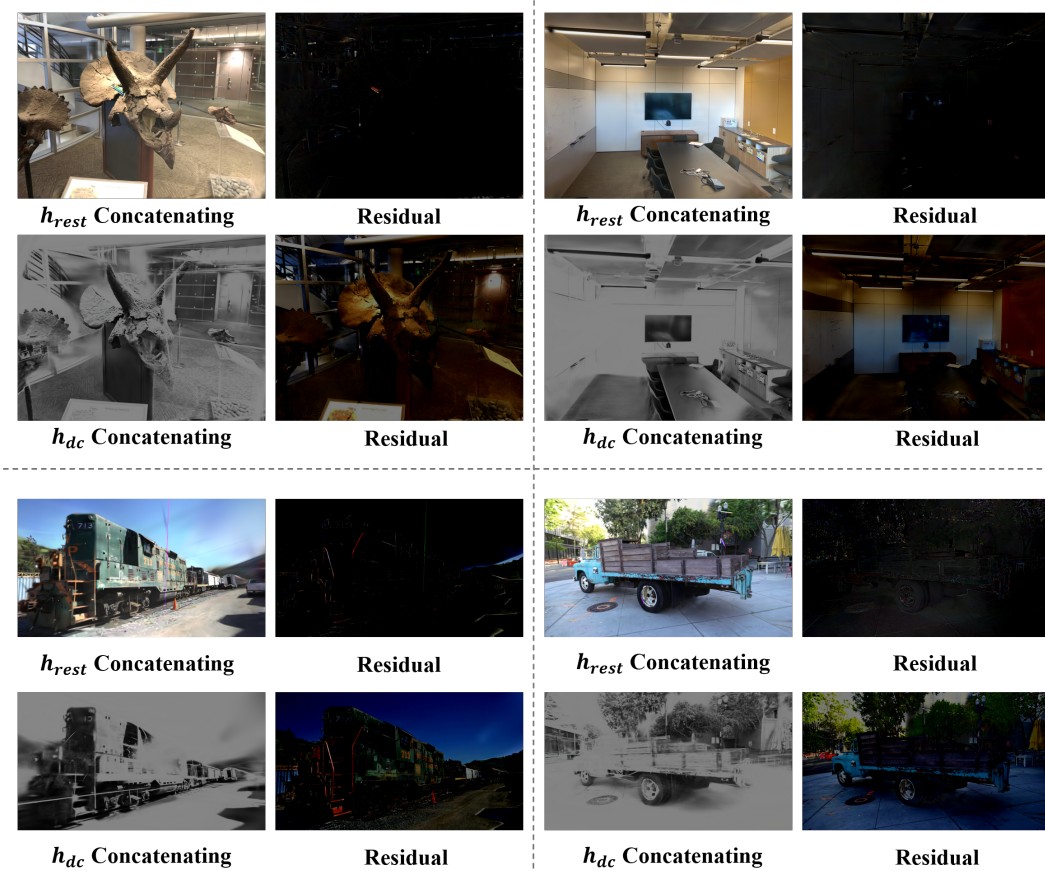

Figure 7: Qualitative comparison of concatenating watermark with different attributes.

### C.2 QUALITATIVE RESULTS FOR VARIOUS EMBEDDING POSITIONS

WATER-GS offers creators the flexibility to fine-tune specific parameters while keeping others fixed. Unlike the method "Concatenating Watermarks with Attributes", our approach does not involve direct value alterations; instead, we fine-tune specific parameters to implicitly embed watermarks. As a result, even when adjusting the same parameters, the images Fig. 10 exhibit significantly smaller distortions compared to those in Fig. 7.

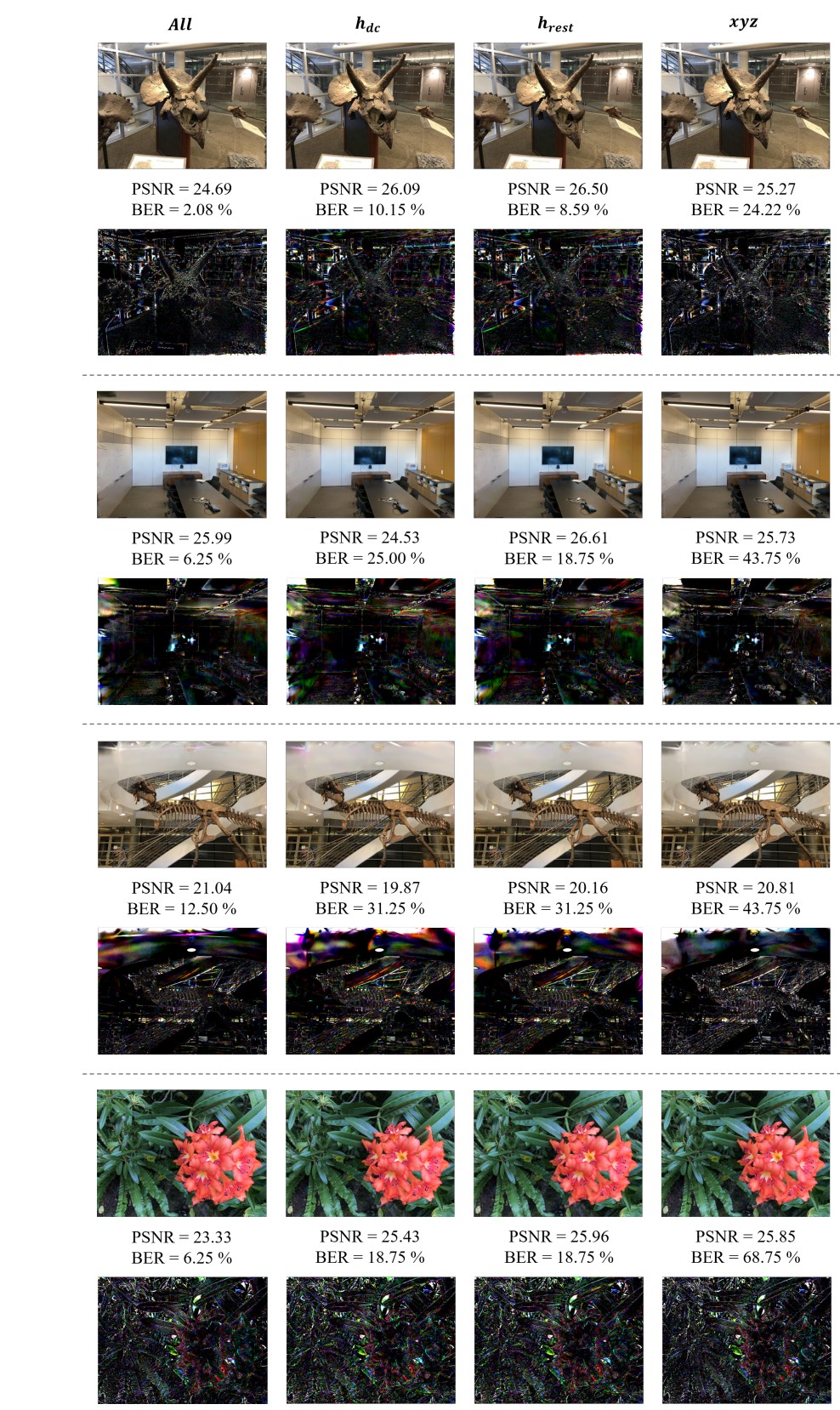

Figure 8: Qualitative comparison of fine-tuning different parameters for watermark embedding.

## C.3 QUALITATIVE RESULTS FOR 2DGS WATERMARK EMBEDDING

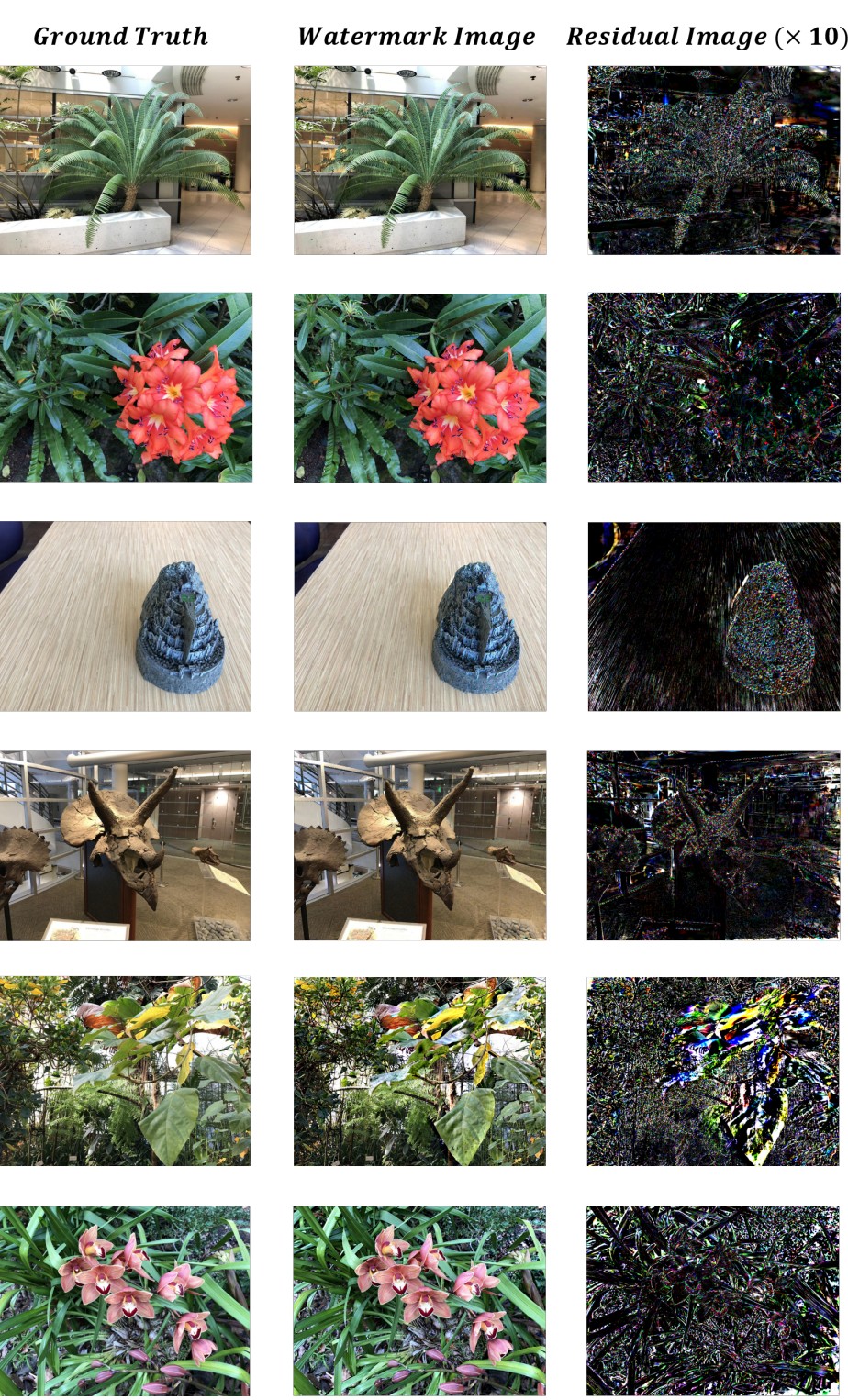

Figure 9: Qualitative comparison of watermark embedding with Compact3D framework. The watermark length is set to 48 bits.

## C.4 QUALITATIVE RESULTS FOR COMPACT3D WATERMARK EMBEDDING

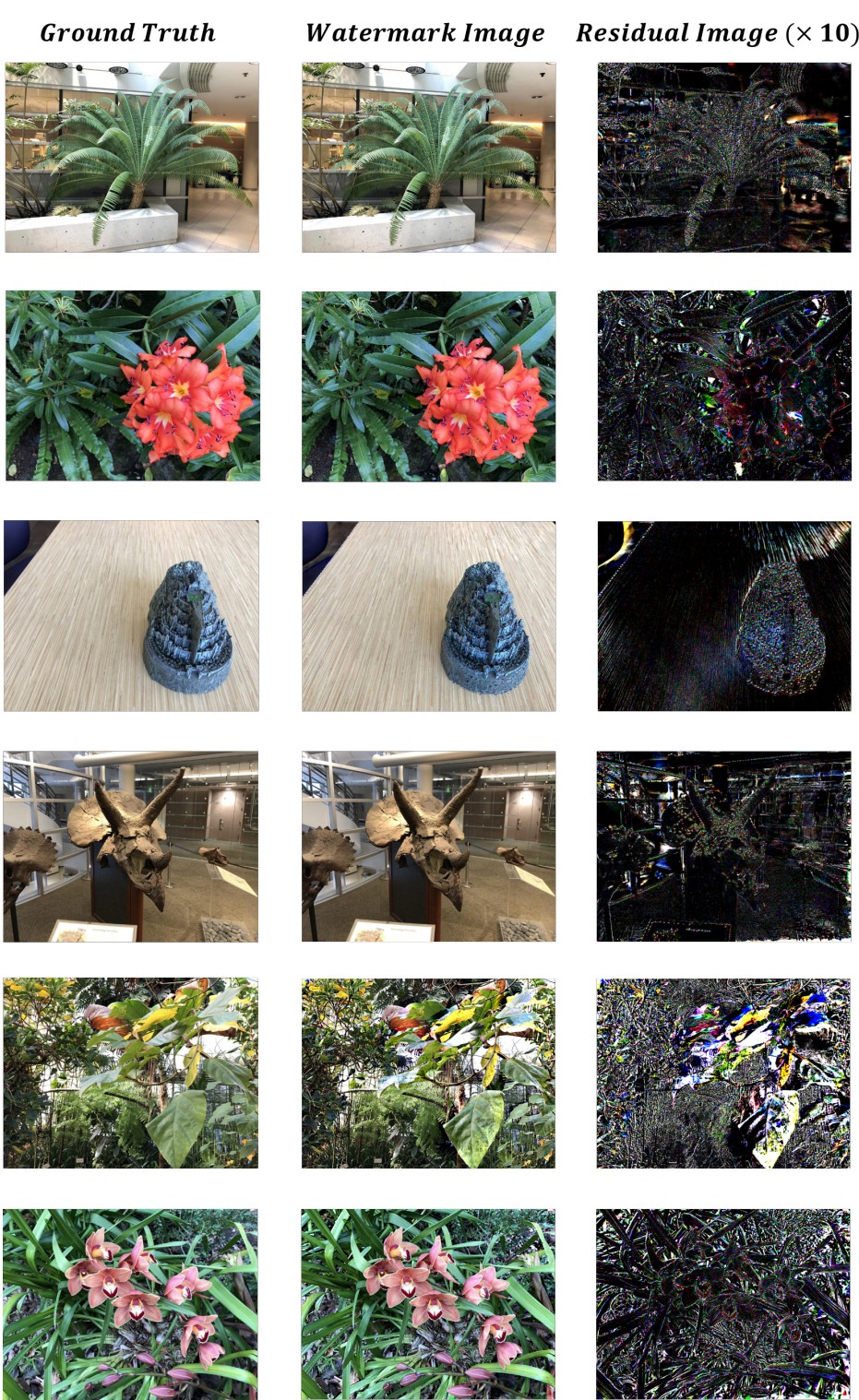

Figure 10: Qualitative comparison of watermark embedding with Compact3D framework. The watermark length is set to 48 bits.

