# OpenReview forum: "WATER-GS: Watermark-embedded 3D Gaussian Splatting via Plug-and-play Decoder"
_ICLR.cc/2025/Conference — ICLR 2025 Conference Withdrawn Submission_

### Official Review · Reviewer_LaGb · 2024-10-29

**Soundness:** 3
**Presentation:** 3
**Contribution:** 1
**Rating:** 3
**Confidence:** 4

**Summary:**

This paper presents WATER-GS, a novel method to protect 3D Gaussian Splatting (3DGS) copyrights via a plug-and-play strategy. WATER-GS uses a pre-trained watermark decoder and innovative 3D distortion layers to embed imperceptible and robust watermarks without compromising rendering efficiency and quality. Experiments show up to a 20% improvement in accuracy, and the method is adaptable to various 3DGS variants.

**Strengths:**

1. The paper is clearly written and well-organized.
2. The experiments effectively demonstrate the method's capabilities.

**Weaknesses:**

1. The novelty of the paper is limited. The proposed method closely resembles NeRFProtector [1] and WateRF [2], but is applied to 3DGS. Such a straightforward adaptation lacks innovation.
2. The baselines used for comparison are too simple. It is essential to include comparisons with NeRFProtector [1] and WateRF [2].
3. There is no evaluation of robustness against 2D distortions, such as clipping and noise.
4. The PSNR values in Table 2, Table 3, and Table 4 are low, which could affect practical usability.

[1] Song, Qi, et al. "Protecting NeRFs' Copyright via Plug-And-Play Watermarking Base Model." arXiv preprint arXiv:2407.07735 (2024).
[2] Jang, Youngdong, et al. "WateRF: Robust Watermarks in Radiance Fields for Protection of Copyrights." Proceedings of the IEEE/CVF Conference on Computer Vision and Pattern Recognition. 2024.

**Questions:**

Please see the Weaknesses.

---

### Official Review · Reviewer_tvLL · 2024-10-29

**Soundness:** 3
**Presentation:** 3
**Contribution:** 2
**Rating:** 3
**Confidence:** 3

**Summary:**

This paper proposes WATER-GS, a novel method designed to protect the copyrights of 3DGS. The method addresses the need to safeguard intellectual property in 3DGS, which has become increasingly important. WATER-GS employs a pre-trained watermark decoder and raw 3DGS generative modules to ensure imperceptibility, along with innovative 3D distortion layers to bolster the watermark's robustness against common distortions. Comprehensive experiments and ablation studies demonstrate  WATER-GS's effectiveness in embedding imperceptible and robust watermarks into 3DGS without sacrificing rendering efficiency or quality. The method is also adaptable to various 3DGS variants, including compression frameworks and 2D Gaussian splatting.

**Strengths:**

- The plug-and-play watermarking module proposed in this paper is simple yet effective.
- This paper provides a comprehensive exploration of the attributes and properties of 3DGS watermarking, with detailed ablation experiments that hold guidance for future work.

**Weaknesses:**

1. This work lacks innovation in terms of technology and methodology. There is no significant difference between this paper and works such as CopyRNerf [1],  WateRF [2], and 3D-GSW [3]. Additionally, the method proposed in this paper has a weak correlation with the nature of 3DGS.
2. The performance of this work is limited. As shown in Table 1, the bit error rate reaches around 9%; the robustness against various attacks is also insufficient, with a significant decrease in bit accuracy under common attacks.
3. This work lacks a comparison with existing 3DGS watermarking methods, such as 3D-GSW [3]. The authors should compare existing 3DGS watermarking methods in the related work or experimental section and demonstrate the advantages of water-gs over these methods.

[1] Luo, Ziyuan, et al. "Copyrnerf: Protecting the copyright of neural radiance fields." Proceedings of the IEEE/CVF International Conference on Computer Vision. 2023.

[2] Jang, Youngdong, et al. "WateRF: Robust Watermarks in Radiance Fields for Protection of Copyrights." Proceedings of the IEEE/CVF Conference on Computer Vision and Pattern Recognition. 2024.

[3] Jang, Youngdong, et al. "3D-GSW: 3D Gaussian Splatting Watermark for Protecting Copyrights in Radiance Fields." arXiv preprint arXiv:2409.13222 (2024).

**Questions:**

Is 48 bit the upper bound of bit watermark capacity for Water-GS, or possibly Water-GS could hide more bits with acceptable visual quality?

---

### Official Review · Reviewer_Eq9x · 2024-11-01

**Soundness:** 2
**Presentation:** 2
**Contribution:** 2
**Rating:** 3
**Confidence:** 5

**Summary:**

This paper introduces WATER-GS, a plug-and-play watermarking method for 3D Gaussian Splatting (3DGS) that embeds imperceptible and robust watermarks using a pre-trained decoder and 3D distortion layers, achieving high extraction accuracy even under real-world distortions.

**Strengths:**

1. WATER-GS is a novel watermarking method for 3D Gaussian Splatting (3DGS) that uses a plug-and-play strategy for easy integration into existing pipelines.
2. A pre-trained watermark decoder is used, treating raw 3DGS generative modules as watermark encoders.
3. Novel 3D distortion layers enhance watermark robustness against real-world distortions of point cloud data.
4. The method is adaptable to different 3DGS variants, including compression frameworks and 2D Gaussian splatting.

**Weaknesses:**

1. The proposed method WATER-GS is very similar to the previous baselines in NeRF such as WaterRF, NeRFProtector, and CopyRNeRF. The proposed WATER-GS basically relies on retuning on the color-related components in CopyRNeRF and relies on the pre-trained decode similar to WateRF and NeRFProtector. Since fine-tuning-based watermarking has been already proven in the radiance field, the contribution of this paper is limited.
2. The evaluation lacks a discussion on security aspects such as watermark security against sophisticated attacks such as adversarial attacks aimed at removing or altering the watermark. Such as PGD perturbations and VAE encoding attacks.
3. The proposed method relies on HiDDeN decoder, which may be vulnerable to distortions such as JPEG compression. More advanced message decoders should be discussed.

**Questions:**

1. The paper mentions 3D distortion layers, are these layers differentiable during the watermark training?
2. How does the self-defined mask select the target parameters? To select the SH parameter in each 3D Gaussian and disable others? Is there any strategy to filter some 3D Gaussians to fine-tune and keep others unchanged?

---

### Official Review · Reviewer_prUe · 2024-11-02

**Soundness:** 2
**Presentation:** 3
**Contribution:** 2
**Rating:** 5
**Confidence:** 4

**Summary:**

This work targets at a challenging and new task: watermarking 3D Gaussian Splatting(3DGS). Authors borrowed the classic end-to-end encoder-decoder watermarking framework as in HiDDeN, treating 3DGS rendering pipeline as the watermark encoder.

**Strengths:**

This paper addresses a vital and challenging task, and it is among the first papers trying to solve this task. The writing and presentation are clear and easy to follow.

**Weaknesses:**

+ W1. **Unclear reasoning of good generation**. Intuitively, watermarking 3DGS would be a much more challenging task than watermarking 2D images, since 3DGS is proposed for *novel-view synthesis*[1]. To protect 3D assets, stakeholders may expect the watermark can be extracted from any-view rendering. Achieving this generalization ability across training/testing views is straightforward for 3DGS rendering (as 3DGS is explicitly modeling 3D physics which should be, by nature view-consistent), however embedding an implicit, imperceptible watermark seems to be not as easy. Intuitively, watermark embedding is not supported by an underlying 3D physics. As shown in the results, the proposed scheme achieved effective generalization from training views to testing views, however readers are not convinced why this generalization could work. Authors should provide more insights into this issue to increase readers' understanding.

+ W2. **Considered distortions are limited**. It seems most of the watermark information is embedded in Spherical Harmonic coefficients (c.f. Table 4). Compared with randomly dropping or cropping Gaussians, a more feasible operation might be transformation of color space: e.g. from Spherical Harmonic representation to RGB space (one example of such color transformation can be referred to https://github.com/graphdeco-inria/gaussian-splatting/issues/485) , for the purpose of aligning with rendering pipeline (some libraries use RGB as color representation) or reducing storage/memory.  Such transformation would completely change the value of color features. Would it break the embedded watermark?

+ W3: **Resilience to reconstruction**. This is more natural to think about compared with "regeneration attacks" in 2D image watermarks, since the original design of 3DGS is to reconstruct 3D from multi-views. Since the 3DGS files are shared online (line 186), if other users try to reconstruct a new 3DGS parameter from the rendering of this watermarked 3DGS file, will the watermark be resilient to such reconstruction?


----------------------------------
**Reference**
+ [1] Kerbl, Bernhard, et al. "3D Gaussian Splatting for Real-Time Radiance Field Rendering." ACM Trans. Graph. 42.4 (2023): 139-1.

**Questions:**

+ Q1. Regarding issues in W1, could authors:
  - Report reconstruction quality and bit error rate on both training views and testing views.
  - Show the distribution of BER across different testing views (instead of only the average results)
  - Discuss if the user wants to randomly select a camera view instead of pre-defined views (this is possible as 3DGS model file uploaded online and can be viewed by a 3D viewer like https://gsplat.tech/), how vulnerable will the watermark be?  Can you provide visualization of views when the watermark extraction breaks?  (I suppose it will be vulnerable since freely choosing camera views could introduce massive rotation, scaling and translation to the training views, and there could be floaters - the under-reconstructed Gaussians. All these factors could hinder watermark extraction).

+ Q2. Regarding W2, could authors
  - Discuss the resilience of proposed watermark to color transformation
  - Show the result of integrating color transformation into their distortion layer

+ Q3. Regarding W3, could authors
  - Report the watermark extraction BER, supposing an adversary train a new 3DGS model from the rendering of watermarked 3DGS model.
  - Discuss potential defenses to defend stronger attacks (e.g. denoising rendered views + reconstruction)

+ Q4. There is a contemporary work[1] also claiming to be the first 3DGS watermarking work. Could authors briefly compare this work with their methods and discuss the strength and weakness compared to [1]?

-------------------------------------
**Reference**
[1] Jang Y, Park H, Yang F, et al. 3D-GSW: 3D Gaussian Splatting Watermark for Protecting Copyrights in Radiance Fields[J]. arXiv preprint arXiv:2409.13222, 2024.

---

### Official Review · Reviewer_1Rny · 2024-11-03

**Soundness:** 3
**Presentation:** 3
**Contribution:** 3
**Rating:** 6
**Confidence:** 5

**Summary:**

This paper presents WATER-GS, a novel plug-and-play approach for robust and imperceptible watermarking in 3D Gaussian Splatting (3DGS) models, achieving high fidelity and resilience against distortions without compromising rendering quality.

**Strengths:**

1. Extensive experiments demonstrate the robustness of WATER-GS under real-world distortions, with up to a 20% improvement in accuracy over other methods, showcasing the method's practical effectiveness in protecting intellectual property.

2. By incorporating 3D distortion layers, the paper enhances watermark robustness, making it resilient against point cloud data distortions and different types of compression, significantly improving over previous techniques.

3. The writing is clear and structured, with thorough explanations of methods, detailed experimental setup, and comprehensive quantitative and qualitative analyses.

**Weaknesses:**

1. The authors omitted some important works on NeRF and GS watermarking [1,2,3] in the related work section. They need to introduce these works and provide a comparison and explanation with them.

[1] 3D-GSW: 3D Gaussian Splatting Watermark for Protecting Copyrights in Radiance Fields.

[2] Gaussianstego: A generalizable stenography pipeline for generative 3d gaussians splatting.

[3] GeometrySticker: Enabling Ownership Claim of Recolorized Neural Radiance Fields.

2. Both Hidden and MBRS struggle to withstand the substantial degradation from 3DGS rendering and training, making the comparison between WATER-GS, 3DGS+Hidden, and 3DGS+MBRS somewhat irrelevant. I recommend that the authors directly apply the WateRF [4] strategy to 3DGS and compare it with WATER-GS, as WateRF is also a plug-and-play watermarking method and is largely independent of the type of 3D representation used.

[4] WateRF: Robust Watermarks in Radiance Fields for Protection of Copyrights.

**Questions:**

Please refer to the weakness.

---

### Official Review · Reviewer_MiHF · 2024-11-05

**Soundness:** 3
**Presentation:** 3
**Contribution:** 2
**Rating:** 3
**Confidence:** 5

**Summary:**

This paper introduces WATER-GS, a novel watermarking method for 3D Gaussian Splatting (3DGS) models to protect their intellectual property. The key idea is using a plug-and-play decoder that treats the original 3DGS generative module as a watermark encoder, allowing creators to embed watermarks without modifying the 3DGS pipeline. The paper also introduces 3D distortion layers to enhance watermark robustness against common point cloud distortions. The method is evaluated on multiple datasets and achieves 95% extraction accuracy while maintaining rendering quality. The approach works with different 3DGS variants including compression frameworks and 2D Gaussian splatting.

**Strengths:**

1. Addresses an important and timely problem of protecting intellectual property for 3DGS models which is becoming increasingly critical as 3DGS gains popularity

2. Practical solution using a plug-and-play design that requires minimal modifications to existing 3DGS pipelines and can work with different variants

3. Comprehensive technical contribution including both the watermark embedding framework and innovative 3D distortion layers to enhance robustness

4. Thorough empirical evaluation with:
- Multiple datasets (LLFF, Mip-NeRF360, Tanks&Temples)
- Extensive ablation studies
- Various distortion scenarios
- Comparison with strong baselines
- Different 3DGS variants

**Weaknesses:**

1. The experimental validation is limited to relatively simple and common scenes. The approach needs to be tested on more challenging scenarios like large-scale environments, dynamic scenes, and real-world applications to demonstrate its practical value and scalability.

2. The high computational cost during the fine-tuning phase (10K-30K iterations taking 0.5-1.5 hours) raises concerns about practical deployment. This overhead may be prohibitive for applications requiring frequent model updates or real-time watermark embedding.

3. The methodology contribution is limited as it is similar to the paper "nerfprotector" (Protecting NeRFs' Copyright via Plug-And-Play Watermarking Base Model), no only by the overall framework, the plug-and-play design but also the writing.

4. The methodology section has gaps in explaining key design choices - particularly the selection of HiDDeN architecture for the decoder and the rationale behind specific parameter settings. More ablation studies on these architectural choices would strengthen the technical contribution.

**Questions:**

Please see Weakness.

---

### Official Review · Reviewer_42VK · 2024-11-06

**Soundness:** 3
**Presentation:** 2
**Contribution:** 2
**Rating:** 5
**Confidence:** 4

**Summary:**

This manuscript proposes a watermarking method in 3D Gaussian Splatting (3DGS) domain.
Fine-tuning of the 3D Gaussians using the pre-trained decoder to embed fixed messages.
The watermarking of 3DGS contains various parameters, including those of ambient and diffuse light in the scene.

**Strengths:**

The manuscript proposes to embed watermarks into 3D Gaussian Splatting while the watermarks can be extracted from the images providing various views of the 3D scene.

**Weaknesses:**

The manuscript is confusing with respect to 3D-based digital watermarking.
In the conclusion it is claimed that the watermarks into variants 3DGS models, which can be accurately extracted from rendered images captured from any viewpoints. However, this important result is not supported during the manuscript either in the methodology or through experiments about how results are extracted from various views of a scene.

There are too few robustness attacks reported in Tables 1 and 3.

Attacks such as JPEG image compression, noise addition and smoothing should have been presented as plots of watermark robustness where the attack is considered at various levels.

**Questions:**

How about changes to the 3D representation, such as point cloud reduction? How this would affect the result of the watermark? The Compact3D compression pipeline is mentioned at the end of the ablation study on page 10, but it is not clear how this attacks works. How much compression of the point cloud is performed?

Is the proposed watermark for 3D watermarking, such as shapes or for 2D images, which is actually seems to be the case? The manuscript mentions several 3D watermarking methods. Why would 3D distortion layers be considered for 2D watermarking.

The results from Tables 1 and 3 including the robustness results seem to be superficial.
They do not include the compression such JPEG compression or smoothing the image.
What sort of crop attacks have been applied? What sort of Gaussian noise level is considered in Table 3?

---

### Note · Authors · 2024-11-14

I have read and agree with the venue's withdrawal policy on behalf of myself and my co-authors.